# Geothermal Assessment of Target Formations Using Recorded Temperature Measurements for the Alberta No. 1 Geothermal Project

**Katherine Y. Huang** [1,*], **Catherine J. Hickson** [1], **Darrell Cotterill** [2] **and Yannick Champollion** [1]

1    Alberta No. 1, Edmonton, AB T5J 3M1, Canada; c.hickson@albertano1.ca (C.J.H.); yannick.champollion@gmail.com (Y.C.)
2    Parallax Resources, Parkland County, AB T7Y 0C4, Canada; parallaxres2014@gmail.com
*    Correspondence: k.huang@albertano1.ca

**Abstract:** The Alberta No. 1 project is a planned power and heat (direct use) geothermal project located within the County of Grande Prairie and Municipal District of Greenview. For the project to successfully produce power and heat on a commercial scale, temperatures of 120 °C are desirable. The produced fluids must also be from highly permeable formations from depths of less than 4500 m. Bottomhole temperature measurements and wireline logs from Alberta's extensive oil and gas database were used to determine the depths to target formations and temperatures within these formations in the project area. The target formations include the dolomitized carbonate units of Devonian age from the Beaverhill Lake Group to the top of the Precambrian Basement. Permeable Devonian-aged sandstone units such as the Granite Wash Formation are also targets. Results suggest that elevation to the top of the Beaverhill Lake Group range from 3104 m to 4094 m and temperatures at the top of the formation range from 87 °C to 123 °C in the study area. Elevation to the top of the Precambrian Basement ranges from 3205 m to 4223 m and temperatures at the formation top range from 74 °C to 124 °C. Within the area where Alberta No. 1 plans to drill, temperatures close to and exceeding 120 °C are expected within the target formations.

**Keywords:** conventional geothermal; direct heat use; Western Canadian Sedimentary Basin; bottomhole temperatures; Alberta

## 1. Introduction

The No. 1 Geothermal Limited partnership (Alberta No. 1) geothermal power and direct heat use project has been awarded funding from Natural Resources Canada's (NR-Can) Emerging Renewable Power Program (ERPP). The program funding matches private sector dollars and stipulates that the geothermal project must produce 5 MWe net of power. To select the project location, Alberta No. 1 conducted a regional study to identify areas in the Alberta portion of the Western Canadian Sedimentary Basin (WCSB) where (1) temperatures are sufficiently high for power production, (2) there are formations at the target depths with known high fluid flows, and (3) there is adequate existing infrastructure that supports low-cost power grid connection as well as direct use applications [1]. Nine chosen areas were assessed for these three constraining factors; results concluded that the area that lies within the Municipal District of Greenview (MDGV) and County of Grande Prairie was most suitable for developing the Alberta No. 1 project (Figure 1).

The study area spans from the northwest corner of Township 73, Range 7, West of the 6th Meridian to the southeast corner of Township 65, Range 3, West of the 6th Meridian (Figure 2). The drilling area, where Alberta No. 1 plans to drill five production and injection wells, spans two ranges and three townships in the vicinity of the Norbord Oriented Strand Board (OSB) facility and a planned light industrial park near the Hamlet of Grovedale

(Figure 2). The park and OSB facility are anticipated industrial heat offtakers from the direct use (heating and cooling) portion of the project.

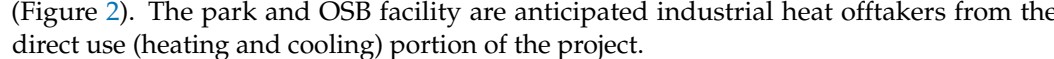

**Figure 1.** The study area lies within the Municipal District of Greenview (MDGV) and the County of Grande Prairie, and encompasses the City of Grande Prairie.

The study area is located within the western portion of the extensive WCSB. In a sedimentary basin such as the WCSB, formations generally decrease in porosity and increase in density (and therefore increase in thermally conductivity) with increasing depth. Permeability and fluid flow are also important parameters when selecting target formations. In the project area, the Precambrian Basement is overlain by thick, permeable carbonate units of Devonian age. Of particular importance are limestone units that have been hydrothermally altered to dolomite creating enhanced permeability [2]. The Alberta No. 1 project will preferentially target these dolomite units and interbedded sandstone units where they are near built infrastructure. Specifically, the target formations span from

the top of the Beaverhill Lake Group to the base of the Granite Wash Formation, which overlies the Precambrian Basement (Figure 3).

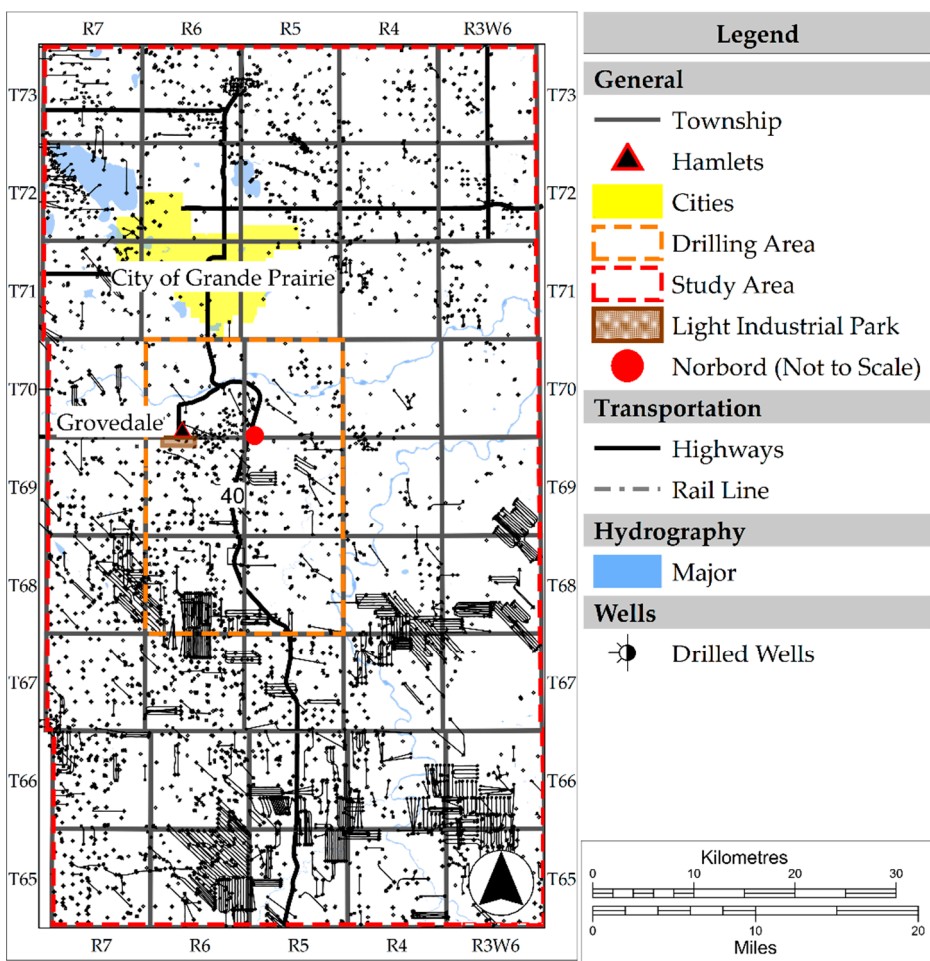

**Figure 2.** The wells previously drilled shown within the study area and the drilling area where Alberta No. 1 plans to drill production and injection wells. The areas encompass a planned light industrial park and the Norbord OSB facility.

To date, over 4000 wells have been drilled in the study area (Figure 2), providing an extensive database to understand the stratigraphy and formation properties of oil and gas targets. However, the data collected differ from the data that would typically be collected for geothermal exploration. The chief difficulty faced by geothermal developers when interpreting bottomhole temperature (BHT) measurements is that the temperatures taken for hydrocarbon development are a perfunctory data point at the end of completion of the well. The data are used for surface engineering designs, especially if the temperatures are high. Wells with BHT data have generally been measured with single, unequilibrated BHT measurements. In comparison, considerable care is taken to obtain accurate and equilibrated temperatures throughout the wellbore for geothermal exploration. This includes the process of allowing the bottom of the well to heat up to thermal equilibrium conditions following drilling. During and after this heat up period, continuous logs are run from top to the bottom. To account for such discrepancies, several correction methods have been created and used to predict equilibrated temperatures at depth from BHTs. Interpretation of geothermal resources from BHT data has been the subject of a considerable amount of research, for example, Harrison et al. (1983), Horner (1951), and Stutz et al. (2012) [4–6].

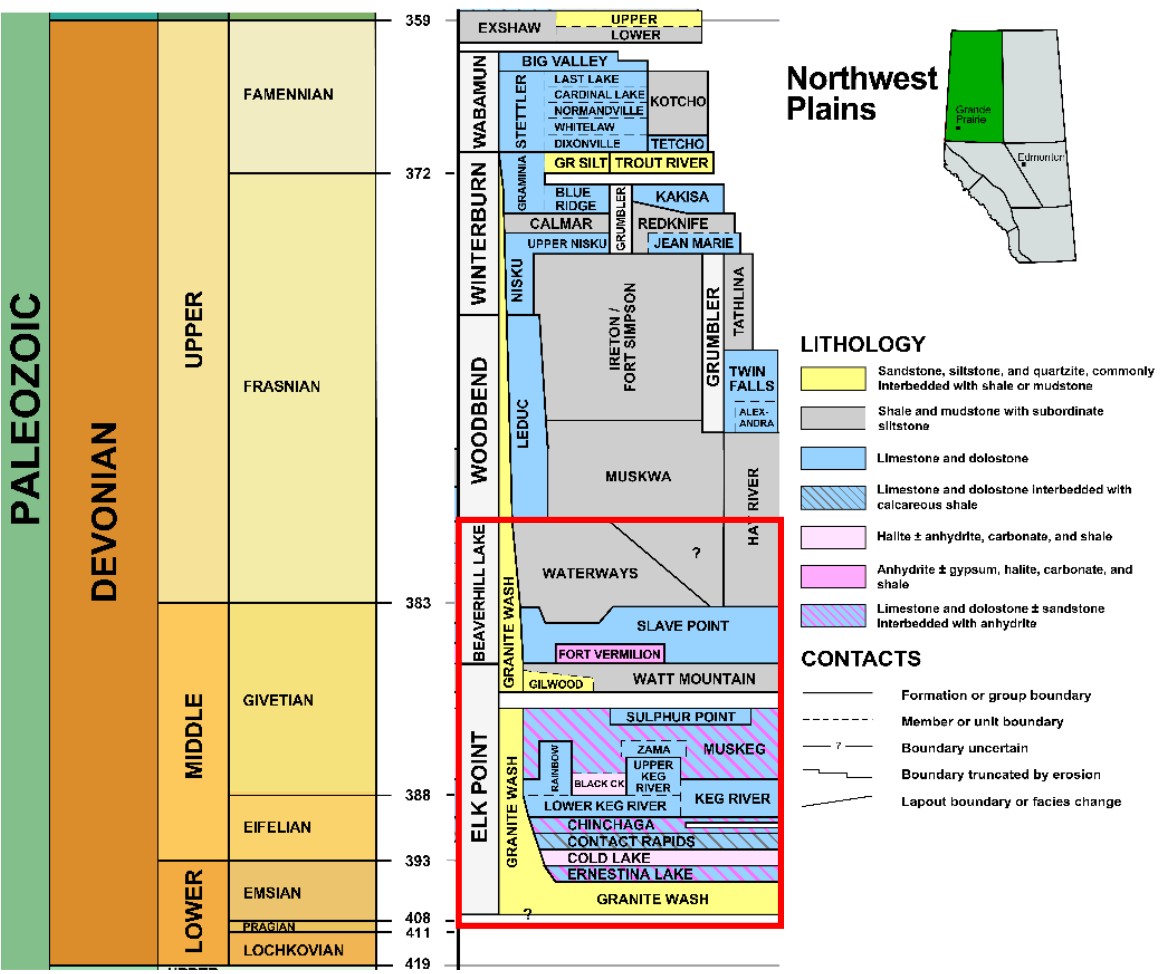

**Figure 3.** General stratigraphy of Devonian units in the Northwest Plains of Alberta. The Alberta No. 1 project area is not underlain by all the units depicted, but is targeting the formations from the Beaverhill Lake Group to the Granite Wash Formation, outlined in red [3].

Previous studies have analyzed oil and gas temperature data specifically within the WCSB to estimate geothermal resources [7–26]. Drill Stem Tests (DST), BHTs, and Annual Pool Pressure (APP) results all include significant errors that require the data to be filtered. BHT measurements generally provide lower temperatures than DST and APP measurements. These studies involve temperature corrections to adjust the recorded temperatures to try to represent the actual temperature at depth.

Several of these correction methods could not be used for this data set; the Horner correction requires input of elapsed time between cessation of circulation in the well bore and the temperature measurement. It is best when several temperature measurements have been made at regular time intervals. The data for this study do not include these measurements, so the Horner correction could not be applied. The thermal gradients of the study areas assessed by Huang et al. (2020) [27] (including the Alberta No. 1 study area) appear to be linear with depth, suggesting that the Harrison correction method, which uses a second-order polynomial fit, is not suitable for the data in this study. Huang et al. (2020) [27] suggest that the filtered, uncorrected BHT data in the WCSB may be more reliable than previously thought. Furthermore, many temperature correction methods increase the temperature from the measured BHT; if this estimation is overly optimistic, the temperature predictions can be detrimental to the project. An overestimation of temperature could falsely suggest that electricity production from the produced fluid is viable, but if the fluid is a few degrees lower, this may not be the case. For the purposes of the Alberta No. 1 geothermal project, a conservative, lower estimate of target formation temperatures is

required for understanding the reservoir, the well design, the expected flow rate, and the power plant design. Therefore, this study uses the raw, filtered BHT data to conservatively predict the lower limit of temperatures at depth. Fluid temperatures of at least 120 °C at depths of 4500 m or less are required to profitably operate the plant. As well, fluid temperature will dictate the required flow rate to produce 5 MWe net.

Our results suggest that elevation to the top of the Beaverhill Lake Group ranges from 3104 m to 4094 m and temperature at the top of the formation ranges from 87 °C to 123 °C in the study area. Elevation to the top of the Precambrian Basement ranges from 3205 m to 4223 m and temperature at the formation top ranges from 74 °C to 124 °C. Within the area where Alberta No. 1 plans to drill, our results calculate depths to the top of the Beaverhill Lake Group and Precambrian Basement to range from 3634 m to 3839 m and 3740 m to 3906 m, respectively. Temperatures range from 87 °C to 123 °C and 89 °C to 127 °C at the top of the Beaverhill Lake Group and Precambrian Basement, respectively.

## 2. Materials and Methods

All available well data, including BHT and True Vertical Depth (TVD) were exported from geoSCOUT from the study area. In total, there were 4261 data points.

### 2.1. BHT Data Filtering and Calculating Thermal Gradient

First, all points that did not include both BHT and TVD data were removed. The average thermal gradient (°C/km) from the surface for each data point was calculated using Equation (1):

$$\text{Thermal Gradient} = 1000 \times \frac{(\text{BHT} - \text{ST})}{\text{TVD}}, \tag{1}$$

where BHT is bottomhole temperature in °C, ST is surface temperature in °C (calculated from mean annual temperature), and TVD is true vertical depth in m. Mean annual temperature of Alberta from 1961 to 1990 was 0.6 °C [28]. The data were then plotted both by BHT vs. depth and thermal gradient vs. depth.

Next, the obvious outliers were removed, including wells with unusually high (>39 °C/km) or low (<20 °C/km) thermal gradients, because they were not consistent with the assumed conductive heat flow in the area and most of the data. The outliers of anomalously high temperatures at high depths were kept for future research, as it may be valuable to look at each data point to assess the legitimacy of the recording.

Other obvious outliers included wells where companies reported the same temperature for multiple wells with different depths. Also, temperature measurements for wells <1 km TVD have been shown to be biased, so these data points were removed [23]. From the work of others, individual outliers of high temperatures at greater depths could be Fahrenheit (F) recorded as Celsius (C), and outlier groups at shallow depth could be due to various factors such as incorrect reading or resetting of the maximum reading thermometers (which give anomalously high temperatures) and, occasionally, recording TVD and BHT as the same value [7]. These errors provide insight into the quality of the data and illustrate that great care must be taken to assess the validity of each measurement

After the filtering process, 1785 data points remained. BHT vs. depth data were plotted then fitted with a linear trendline to calculate the averaged uncorrected thermal gradient. The thermal gradient vs. depth data were also plotted and fitted with a linear trendline to assess the change in thermal gradient with depth.

The average thermal gradient of these wells was calculated to be 23.9 °C/km (Figure 4). The average thermal gradient change with depth was calculated to be −1.5 °C/km, suggesting that gradient does not change significantly with depth. This means that using the gradient to calculate temperature at depths may be straightforward, but caution should be taken when interpreting the results and extrapolating to depth due to uncertainty of the accuracy of the BHT data.

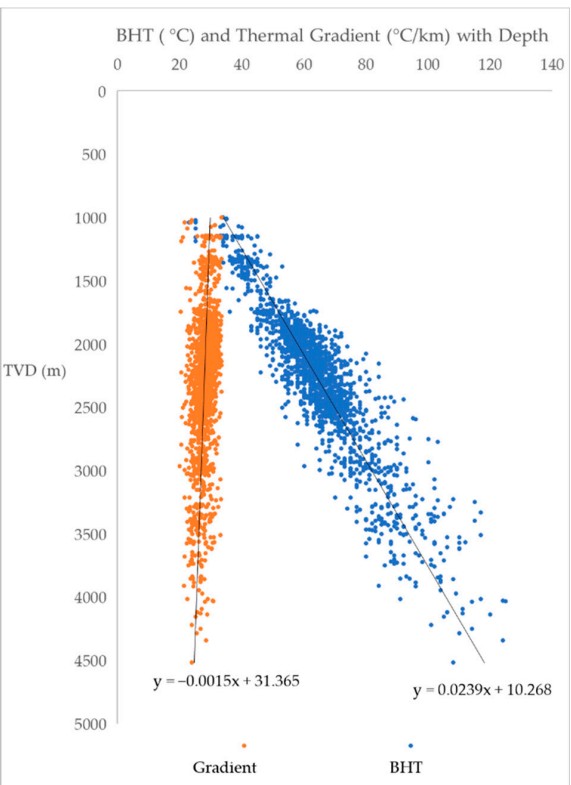

**Figure 4.** The calculated thermal gradient with depth of each well is shown as orange points. The BHT with depth of each well is shown as blue points.

### 2.2. Selecting Formation Tops

After the data were filtered, depths to target formations were analyzed. First, data points from horizontal wells, as well as re-entered well events, were removed. Additionally, six wells did not have wireline logs. In total, 1655 wells were assessed for formation tops. Depths to all formations between the Beaverhill Lake Group and the top of the Precambrian Basement were chosen by assessing wireline logs of each well that drilled to these formations and each TVD recorded. The wireline logs used were resistivity, gamma, neutron-density, and sonic if the former three logs were unavailable.

TVD data for all formation tops for each well were imported into Surfer. Next, grid files were created from TVD data for each formation and used to create contours with a simple Kriging method. Within the drilling area, only 2 wells penetrate to the Beaverhill Lake Formation and deeper. Therefore, the elevation contour maps were extrapolated to calculate the expected depth to Beaverhill Lake Group and the Precambrian Basement from the 16 wells that penetrate to the Wabamun Formation in the drilling area. This was done using the Point Sample calculator on Surfer.

### 2.3. Temperature and Elevation Maps

After formation tops were selected and depths to the Beaverhill Lake Group and the Precambrian Basement were extrapolated, we calculated the expected temperature at these formation tops using Equation (2):

$$T_{Formation} = \frac{\text{Thermal Gradient} \times \text{Formation Top TVD}}{1000}, \tag{2}$$

where $T_{Formation}$ is the temperature at the top of the formation at each well in °C, and Formation Top TVD is the depth to the top of the formation in m. For the study area, we used the TVDs from the well data. For the drilling area, we used the TVDs calculated from the elevation contour maps.

T_{Formation} data for each well were imported into Surfer. Next, grid files were created from the temperature data for both formations and used to create contours with a simple Kriging method.

## 3. Results

Within the study area, the depth to the top of the Beaverhill Lake Group ranges from 3104 m to 4094 m (Figure 5a) and temperature at the top ranges from 72 °C to 123 °C (Figure 5b). The depth to the top of the Precambrian Basement ranges from 3205 m to 4223 m (Figure 6a) and temperature at the top ranges from 74 °C to 124 °C (Figure 6b).

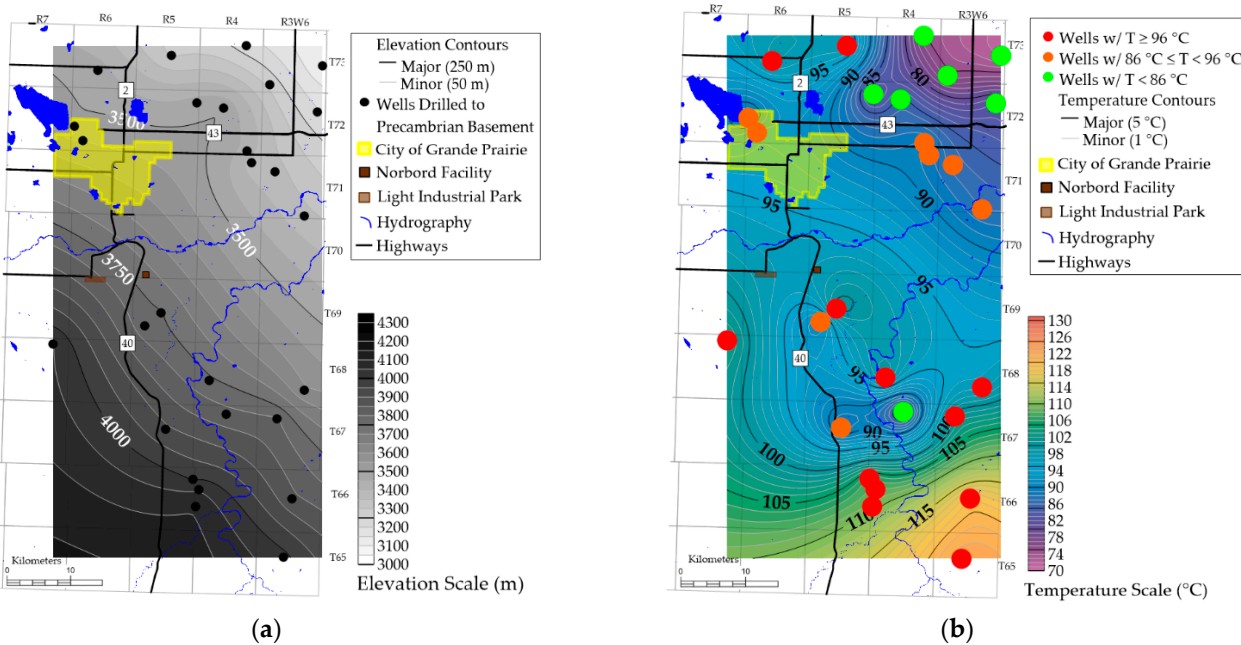

**Figure 5.** Contour maps of the study area for the Beaverhill Lake Group showing (**a**) depth to the top and (**b**) temperature at the top.

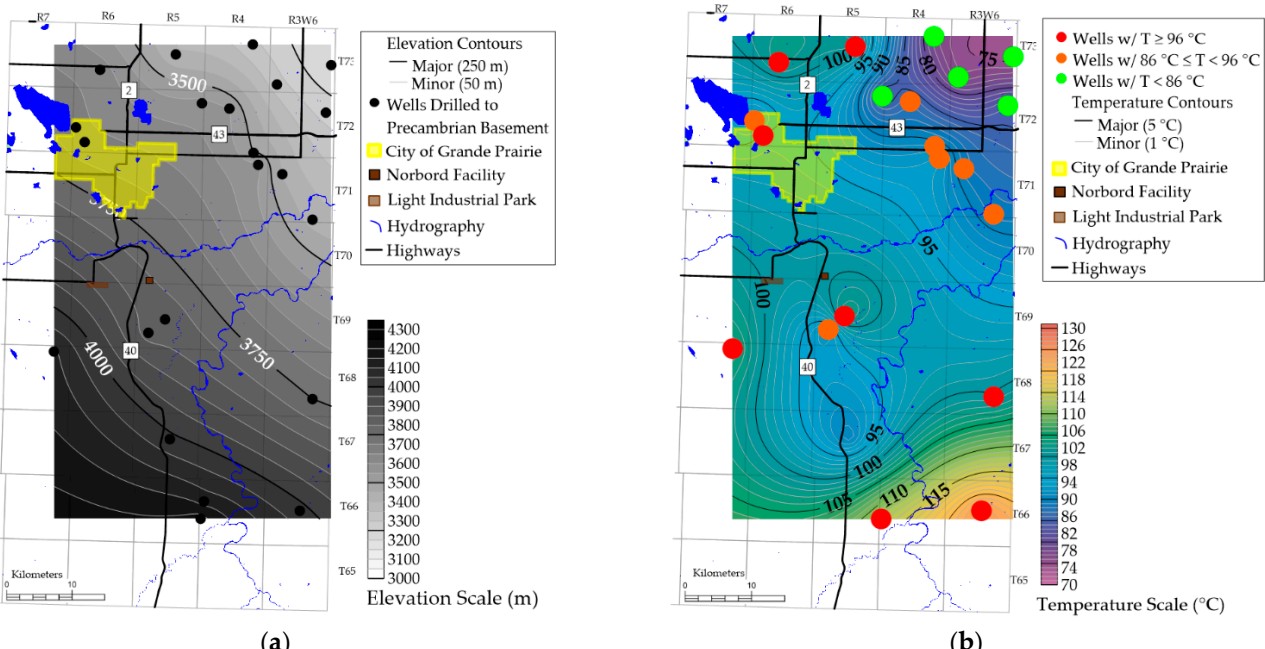

**Figure 6.** Contour maps of the study area for the Precambrian Basement showing (**a**) depth to the top and (**b**) temperature at the top.

Within the drilling area, the depth to the top of the Beaverhill Lake Group ranges from 3634 m to 3839 m (Figure 7a) and temperature at the top ranges from 87 °C to 123 °C (Figure 7b). The depth to the top of the Precambrian Basement ranges from 3740 m to 3906 m (Figure 8a) and temperature at the top ranges from 89 °C to 127 °C (Figure 8b).

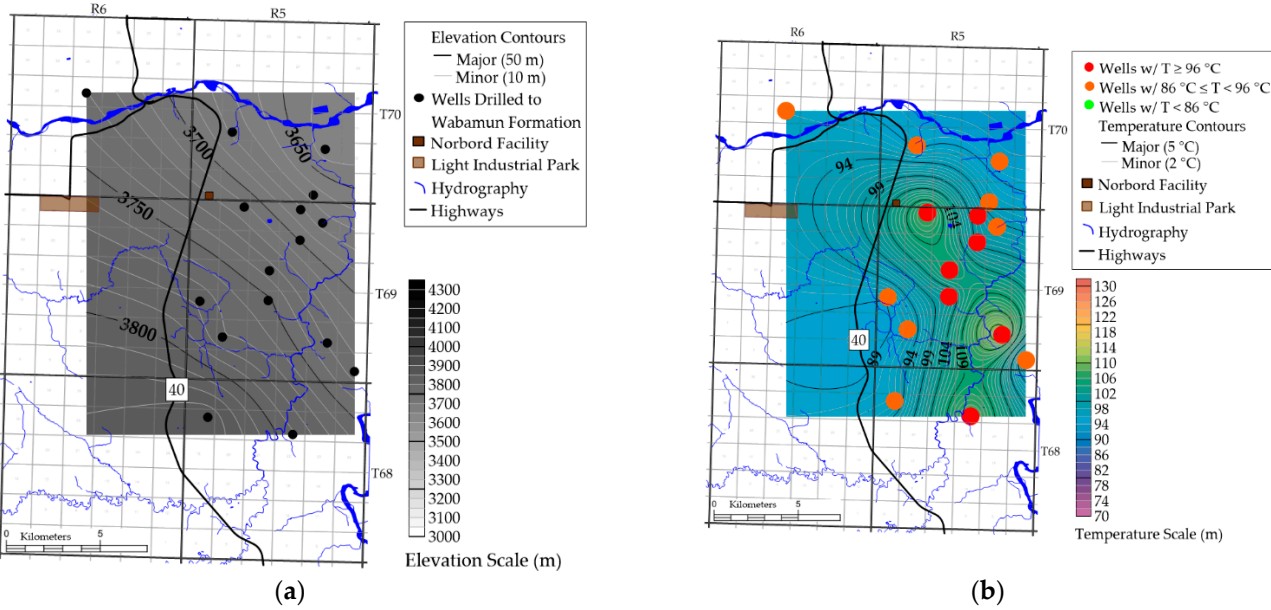

**Figure 7.** Contour maps of the drilling area for the Beaverhill Lake Group showing (**a**) depth to the top and (**b**) temperature at the top.

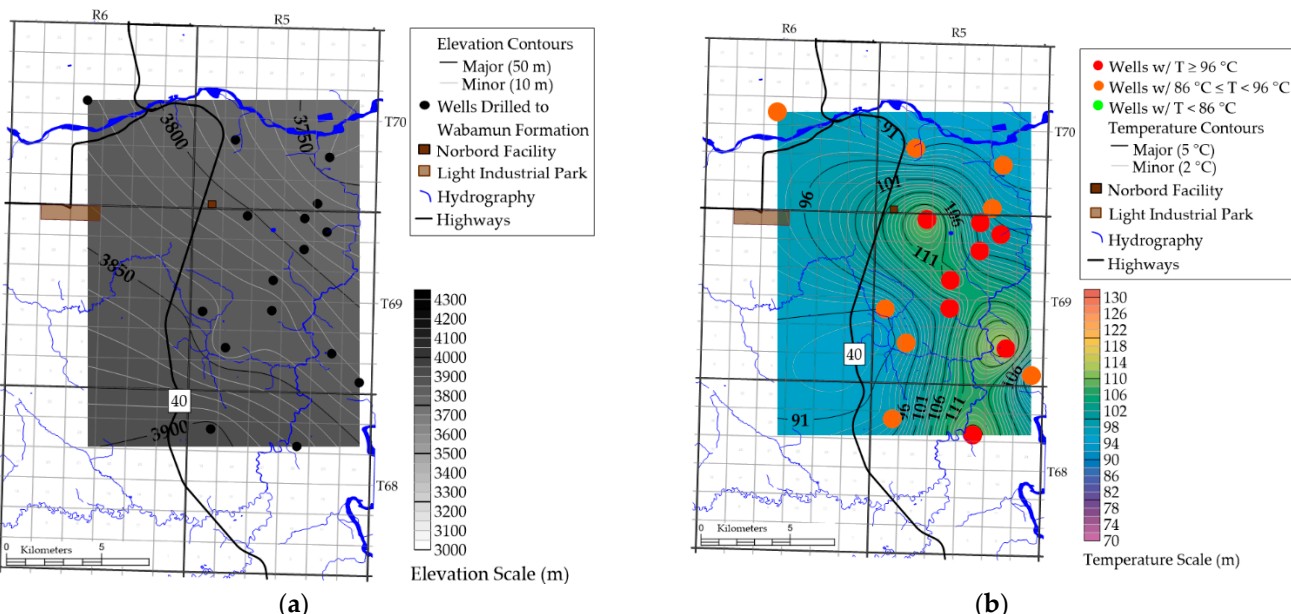

**Figure 8.** Contour maps of the drilling area for the Precambrian Basement showing (**a**) depth to the top and (**b**) temperature at the top.

On both a regional and local scale, the Beaverhill Lake Group and the Precambrian Basement both appear to be dipping to the southwest. From the limited data set, there are no apparent structural features that may indicate faulting through the formation tops.

Based on the temperature maps, there is a rough correlation between the southwest dipping stratigraphy and increasing temperature within the study area. This correlation,

however, is not present in the drilling area, which may be due to the scarce data set. There does appear to be a minor southeast- and south-trending temperature high which may be caused by a structural feature in this direction. However, this data set is too scarce to confidently determine the existence of this structure.

Because the temperature data is not corrected, great care must be taken when interpreting the results. The oil and gas data do not include fluid circulation time during drilling; therefore, it is unknown if BHTs represent equilibrium temperature. The actual formation temperatures will not be known until the first well is drilled and logged using proper equipment and methods. The temperatures from this study provide a conservative estimate for the purposes of exploration.

### 4. Conclusions

For a geothermal project to be economic in the area at this time, the estimated maximum depth of target formations must be less than 4500 m, and thermal fluids of at least 120 °C provide the best opportunity for commercial viability. The Devonian carbonate and sandstone formations between the Beaverhill Lake Group and the Precambrian basement are less than 4500 m depth in the study area. Within the drilling area, there are sections where temperatures have been calculated to almost reach or exceed 120 °C to the east and southeast of the Norbord OSB plant. Because we expect the BHT data to provide a lower-end temperature estimate, the actual formation temperatures may be higher. Based on these results, the Alberta No. 1 geothermal power and direct use heat project in this area fits the depth and temperature criteria to be successful. Other geological considerations that will be assessed include flow rate potential of these formations and areas of fluid convection such as faults and fractures, which can be delineated by seismic data.

**Author Contributions:** Conceptualization, C.J.H. and K.Y.H.; Methodology, K.Y.H., C.J.H., D.C. and Y.C.; Software, K.Y.H. and D.C.; Validation, K.Y.H., C.J.H., D.C. and Y.C.; Formal Analysis, K.Y.H. and D.C.; Investigation, K.Y.H. and D.C.; Resources, K.Y.H. and D.C.; Data Curation, K.Y.H. and D.C.; Writing—Original Draft Preparation, K.Y.H.; Writing—Review & Editing, K.Y.H. and C.J.H.; Visualization, K.Y.H.; Supervision, C.J.H.; Project Administration, K.Y.H. and C.J.H. All authors have read and agreed to the published version of the manuscript.

**Funding:** This research received no external funding.

**Informed Consent Statement:** Not Applicable.

**Data Availability Statement:** Data was obtained from the geologic systems' geoSCOUT program. geoSCOUT is available at https://www.geologic.com/products/geoscout/ on a subscription basis.

**Acknowledgments:** This work was carried out by the Alberta No. 1 team supported by Terrapin Geothermics. The Alberta No. 1 project is partially funded by Canada's federal government, through Natural Resources Canada's Emerging Renewable Power Program. The strong support of the Federal government for this project is gratefully acknowledged. The input and discussions on subsurface temperatures with Will Gosnold and Dick Benoit are sincerely appreciated.

**Conflicts of Interest:** The authors declare no conflict of interest.

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
