# Peer review of "Geothermal Assessment of Target Formations Using Recorded Temperature Measurements for the Alberta No. 1 Geothermal Project"

_applsci, doi:10.3390/app11020608_

Round 1

Reviewer 1 Report

Introduction/discussion can be improved by addressing similar work and compare your methodology and results with. Also highlight the importance of the presented data for planning geothermal projects.

Some examples are:

Bonté, Damien, J-D. Van Wees, and J. M. Verweij. "Subsurface temperature of the onshore Netherlands: new temperature dataset and modelling." Netherlands Journal of Geosciences 91.4 (2012)

Willems, C. J. L.,.. "Towards optimisation of geothermal heat recovery: An example from the West Netherlands Basin." Applied energy 247 (2019).

Author Response

Thank you for your review and comments.

  1. Introduction/discussion can be improved by addressing similar work and compare your methodology and results with.

This has been addressed in the introduction. Methodology of studies within the WCSB have been briefly outlined and we have explained why we did not use this methodology

  1. Also highlight the importance of the presented data for planning geothermal projects.

This has been addressed in the introduction

Thank you for the examples. The papers (Bonté et al., 2012; Willems, 2019) have been reviewed but were not referenced in the paper.

Reviewer 2 Report

the last two authors affiliation is not listed.

it would be helpful if the drilling area was highlighted on the study area map

figure 3 requires at least 200% zoom to be readable; same for Figure 6

almost impossible to tell difference between red and orange wells in Figure 7b, 8b

Author Response

Reviewer #2

Thank you for your review and comments.

The following changes have been made:

  1. The last two authors affiliation is not listed.

Affiliation has been updated

  1. It would be helpful if the drilling area was highlighted on the study area map

Drilling area has been highlighted on both study area map and project map (Figure 1)

  1. Figure 3 requires at least 200% zoom to be readable; same for Figure 6

Figure 3 has been replaced. Figures 6-9 have been updated with increased font size

  1. Almost impossible to tell difference between red and orange wells in Figure 7b, 8b

Size of symbols has been increased for easier distinction

Reviewer 3 Report

This manuscript describes the results of the assessment of the temperatures of the potential geothermal target formations of Devonian age selected in County of Grande Prairie and Municipal District of Greenview. The aim is to use geothermal source with a temperature close to and exceeding 120°C for power and heat production. The authors describe a methodology of using bottomhole temperature measurements to predict and map the temperature at the interested geological surfaces at a depth of up to 4500 m.

The authors use the simplest method based solely on a geothermal gradient calculation, what seems to be insufficient compared to the work that have been tackled by previous authors, e.g. Weides, S., Majorowicz, J., (2014). This also applies to the extensive series of publications of: Majorowicz's, Jessop's, Grasby's and others. Additionally, the methodology used requires some clarification, as indicated in the PDF file attached.

Going into details it was explained that corrections to raw BHT maeasurements have been made but in the study the uncorrected data were used. This requires further explanation, the more that the corrected input data should, in principle, provide better quality and better results at last.

The methodology used to plot temperature maps by 1D interpreatation of the data, calculated separately in particular borehole, doesn't guarantee that the obtained map series not intersect each other. Better results can only be achieved when working on grids. This may be particularly important in the case of low area coverage with boreholes and small thickness between the analyzed horizons. It is also recommended to check this by simply subtracting the obtained temperature maps (grids) - if not done yet?

Some of the figures included doesn't provide any significant information, thus please consider their removal or to merge some of them if aplicable. Please also improve the overall readability of the maps by enlargement the descriptions etc.

The sections authors’ contribution, as well as conflict of interest is missing.

The overall contribution of the paper to scientific knowledge is rather low.

Detailed comments can be found in the attached PDF file.

Reference cited:

Weides, S., Majorowicz, J., 2014. Implications of Spatial Variability in Heat Flow for Geothermal Resource Evaluation in Large Foreland Basins: The Case of the Western Canada Sedimentary Basin. Energies, 7(4), 2573-2594; https://doi.org/10.3390/en7042573

Author Response

Reviewer # 3

Thank you very much for reviewing our paper thoroughly and providing your insights. They are all appreciated and taken into consideration. We believe that your comments have improved the overall quality of the paper.

  1. The authors use the simplest method based solely on a geothermal gradient calculation, what seems to be insufficient compared to the work that have been tackled by previous authors, e.g. Weides, S., Majorowicz, J., (2014). This also applies to theextensive series of publications of: Majorowicz's, Jessop's, Grasby's and others. Additionally, the methodology used requires some clarification, as indicated in the PDF file attached.

Regarding the comments on the methodology and content of paper itself- the purpose of this paper is not to disregard the important work done by Majorowicz, Weides, Grasby, and others. Rather, the purpose is to provide a conservative estimate of temperature within target formations for the commercial development of the Alberta No. 1 power and direct heat geothermal project. As stated in the paper, most of the corrections applied by the above-named authors increase the temperature from the BHT measured from oil and gas projects. If this estimation is overly optimistic, the temperature predictions can be detrimental to the project, especially at temperatures that are within the margin of the fluids being able to produce electricity. We have also included in the introduction why some of these correction methods were not applied to our data.

  1. Going into details it was explained that corrections to raw BHT maeasurements have been made but in the study the uncorrected data were used. This requires further explanation, the more that the corrected input data should, in principle, provide better quality and better results at last.

Addressed in point #12.

  1. The methodology used to plot temperature maps by 1D interpreatation of the data, calculated separately in particular borehole, doesn't guarantee that the obtained map series not intersect each other.Better results can only be achieved when working on grids. This may be particularly important in the case of low area coverage with boreholes and small thickness between the analyzed horizons. It is also recommended to check this by simply subtracting the obtained temperature maps (grids) - if not done yet?

Addressed in point #16

  1. Some of the figures included doesn't provide any significant information, thus please consider their removal or to merge some of them if aplicable.Please also improve the overall readability of the maps by enlargement the descriptions etc.

Addressed in points #8, #9, #10, #11

  1. The sections authors’ contribution, as well as conflict of interest is missing.

This paper was not written in sections by authors. We do not believe there is conflict of interest to note.

  1. The overall contribution of the paper to scientific knowledge is rather low.

  1. Convention of space before units

This has been resolved

  1. Figure 1: Scale of overview map and enlarge text on legend

Scale of overview map has been changed, text on legend has been enlarged

  1. Figure 2: Merge both maps, improve readability

We feel that we did not make the purpose of the “Study Area” and the “Drilling Area” clear enough; we have updated these figures into Figure 2, as well as included the “Drilling Area” into Figure 1. The font size has been increased

  1. Figure 3: Not much information about geology, transfer relevant information to a previous map, show study area on map

Figure 3 was removed.

  1. Figure 4: indicate target aquifer boundaries, increase readability

This is now Figure 3. Target formations have been outlined. Figure size has increased and font is readable at 200%

  1. Introduction: Explain why uncorrected, raw BHT measurements are used

This has been added to the introduction

  1. Numbering of formulas

This has been updated

  1. Calculated gradient in text does not match graph. Considering the plot attached the annual mean surface temperature the study area equal to ca. 10C – is it right? The use of such a simplified model based on raw data implies many uncertainties- difficult to verify and interpret

Gradient has been updated in text.

Uncertainties when using single BHT and depth point are addressed in the introduction. Uncertainty when interpreting results is addressed in the discussion

  1. Equation 2: Average ambient temperature. A well known formula comprise also To that as follows: Tf = To + (Grad T x TVD). In your formula the component To is missing, isn’t it? Please consider the Equation no (1)

Annual mean surface temperature is included in Equation 1 (as ST)

  1. Results: Did you check the possible intersection of the obtained map series? Using the proposed methodology, the intersection of temperature maps will most likely occur

This is beyond the scope of the paper. Estimating the temperature at the top of the uppermost target formation and the lowermost target formation is within the scope of the information required for development for the Alberta No. 1 project.

  1. Figure 6: Scale, larger font size

Scales have been added and font size increased for Figures 6-9

  1. Conclusions: The analysis of the map shows that the expected temperature anomaly above 120 C is located about 15km SE of the potential energy user. Lack of information on the possible error in performing the analysis and the obtained results doesn’t allow for a decision whether the assumed temperature threshold could be achieved or not

Possible error has been addressed in the discussion. Ultimately, the actual formation temperatures, as well as the suitability of correction methods applied in previous studies, will not be known until a geothermal well is drilled to target depths and temperature logged.

Round 2

Reviewer 2 Report

None

Author Response

The revised manuscript is attached.

Reviewer 3 Report

  • I accept most of the amendments made, except for the ones below:
  • Please double check through the text the space convention in front of the units.
  • All comments addressed to points no. 14, 15 can be found below point no 15 (with yellow backgroud).
  1. Calculated gradient in text does not match graph. Considering the plot attached the annual mean surface temperature the study area equal to ca. 10C – is it right? The use of such a simplified model based on raw data implies many uncertainties- difficult to verify and interpret

Gradient has been updated in text.

Uncertainties when using single BHT and depth point are addressed in the introduction. Uncertainty when interpreting results is addressed in the discussion

  1. Equation 2: Average ambient temperature. A well known formula comprise also To that as follows: Tf = To + (Grad T x TVD). In your formula the component To is missing, isn’t it? Please consider the Equation no (1)

Annual mean surface temperature is included in Equation 1 (as ST)

The basic concern in the paper consider the use of real data (annual mean temperature) to a priori fake temperature distribution model, based on uncorrected BHT measurements. On the one hand, to evaluate the gradients in particular boreholes, you used to use the real temperature ST=0.6, which has nothing to do with the temperature ST resulting from the adopted model (see Fig. 4). The figure shows that the average temperature "ST" derived from the model is equal to 10.268 degC, and the gradients for individual borehole and measurements should be calculated according to the formula: ???????? = 1000 × (??? − 10.268) / ???. Honestly speaking "ST" should express a temperature within the thermally neutral zone, ca. 20 m below the surface - (not the ambient temperature). Furthermore in equation (2) the "ST" parameter doesn't appear! The transformation of formula (1) yields another formula, namely: ?????????? = ???????? × ????????? ??? ??? / 1000 + ST. The "ST" parameter disappeared from your formula - why? Please see the attached xlsx file (green columns: X, Y, AB have been added).

Please find a solution to the problem.

  1. Results: Did you check the possible intersection of the obtained map series? Using the proposed methodology, the intersection of temperature maps will most likely occur

This is beyond the scope of the paper. Estimating the temperature at the top of the uppermost target formation and the lowermost target formation is within the scope of the information required for development for the Alberta No. 1 project.

  • To avoid further questions about possible intersection of the temperature map set, please consider drawing contour maps with a step of 5 degC, this would partly resolve the problem. At Figure 7 the legend doesn't fit the map content (isolines value).
